# Comparative Analysis of ROS Network Genes in Extremophile Eukaryotes

**DOI:** 10.3390/ijms21239131

**Published:** 2020-11-30

**Authors:** Rafe Lyall, Zoran Nikoloski, Tsanko Gechev

**Affiliations:** 1Department Bioinformatics and Mathematical Modelling, Center of Plant Systems Biology and Biotechnology, 139 Ruski Blvd., 4000 Plovdiv, Bulgaria; Nikoloski@mpimp-golm.mpg.de (Z.N.); gechev@cpsbb.eu (T.G.); 2Bioinformatics, Institute of Biochemistry and Biology, University of Potsdam, Karl-Liebknecht-Str. 24-25, 14476 Potsdam-Golm, Germany; 3Systems Biology and Mathematical Modelling Group, Max Planck Institute of Molecular Plant Physiology, Am Mühlenberg 1, 14476 Potsdam-Golm, Germany; 4Department of Plant Physiology and Molecular Biology, Plovdiv University, 24 Tsar Assen str., 4000 Plovdiv, Bulgaria

**Keywords:** ROS, extremotolerance, resurrection plants

## Abstract

The reactive oxygen species (ROS) gene network, consisting of both ROS-generating and detoxifying enzymes, adjusts ROS levels in response to various stimuli. We performed a cross-kingdom comparison of ROS gene networks to investigate how they have evolved across all Eukaryotes, including protists, fungi, plants and animals. We included the genomes of 16 extremotolerant Eukaryotes to gain insight into ROS gene evolution in organisms that experience extreme stress conditions. Our analysis focused on ROS genes found in all Eukaryotes (such as catalases, superoxide dismutases, glutathione reductases, peroxidases and glutathione peroxidase/peroxiredoxins) as well as those specific to certain groups, such as ascorbate peroxidases, dehydroascorbate/monodehydroascorbate reductases in plants and other photosynthetic organisms. ROS-producing NADPH oxidases (NOX) were found in most multicellular organisms, although several NOX-like genes were identified in unicellular or filamentous species. However, despite the extreme conditions experienced by extremophile species, we found no evidence for expansion of ROS-related gene families in these species compared to other Eukaryotes. Tardigrades and rotifers do show ROS gene expansions that could be related to their extreme lifestyles, although a high rate of lineage-specific horizontal gene transfer events, coupled with recent tetraploidy in rotifers, could explain this observation. This suggests that the basal Eukaryotic ROS scavenging systems are sufficient to maintain ROS homeostasis even under the most extreme conditions.

## 1. Introduction

The proliferation of early photosynthetic organisms nearly 2.5 billion years ago led to a massive increase in atmospheric oxygen on early Earth [1]. Surviving in the presence of oxygen has thus irrevocably shaped the evolution of life on the planet. The sequential reduction of atmospheric dioxygen produces highly reactive oxygen radicals collectively known as reactive oxygen species (ROS). ROS encompass a broad range of molecules that display varying reactivity, cellular mobility and half-lives [2,3,4]. Many ROS can react with cellular components, such as proteins, lipids and nucleic acids, leading to metabolic dysregulation and cell death. The sensing and maintenance of noncytotoxic ROS levels in early organisms required a complex network of ROS-related genes, particularly ROS scavengers. In complex organisms these systems also evolved to sense perturbations in basal ROS levels—depending on the origin or species of ROS molecules produced—enabling the use of ROS as signaling molecules [5,6]. All aerobic organisms, and some anaerobic ones, likely encounter oxidative stress on a regular basis. An imbalance in ROS homeostasis can lead to oxidative stress, resulting in direct cellular damage that is exacerbated by a breakdown in ROS signaling mechanisms. Organisms that encounter extreme abiotic stress conditions—such as extremophiles—may need to tolerate far more drastic perturbations to ROS homoeostasis than species in more moderate environments. However, it is unclear whether the effects of this lifestyle would necessitate changes to the core ROS-related gene networks. The goal of this study was to analyze ROS-related genes in the genomes of diverse Eukaryotes to see if the extremophile lifestyle was associated with changes to any of the core ROS-related gene families.

ROS are continuously produced during cellular metabolism and through the action of ROS-generating oxidases, and subsequently detoxified through a diverse array of ROS scavenging mechanisms (Table 1). Nonenzymatic antioxidants are sacrificial molecules that will readily scavenge specific forms of ROS, acting as an electron donor and becoming oxidized in the process [7]. Some common antioxidants include ascorbic acid (vitamin C; AsA), all-trans retinol (vitamin A), alpha-tocopherol (vitamin E), beta-carotene and glutathione (GSH) [8]. GSH and AsA function primarily as cofactors to enzymatic antioxidants in the respective GSH/AsA cycles (Figure 1) [9,10,11]. Another major source of downstream ROS occurs through the reaction of metal ions with hydrogen peroxide (H_2_O_2_) to produce the unstable hydroxyl radical, particularly via Haber–Weiss and Fenton-type reactions [12]. Metal chelating enzymes, such as ferritin, can thus also act as antioxidants by their capacity to sequester metal ions from ROS and prevent the production of more dangerous radical species [13]. Enzymatic ROS scavengers detoxify ROS directly or by using antioxidant cofactors. Some enzymatic ROS scavengers, such as superoxide dismutases (SOD) or catalases (CAT), act as the first line of defense against specific ROS species [14]; others, like the enzymes involved in the GSH redox cycle (glutathione peroxidase (GPX) and peroxiredoxin (Prxr), glutaredoxins/thioredoxins (GRX/TRX) and glutathione reductase (GR)) and the photosynthetic AsA-GSH cycle (ascorbate peroxidase (APX), monodehydroascorbate reductase (MDHAR) and dehydroascorbate reductase (DHAR)), are vital in maintaining the pools of reduced antioxidants required to protect the cell from ROS (Figure 1) [9,15,16]. The ROS-generating family of NADPH oxidase (NOX) enzymes are found almost exclusively in multicellular organisms and function purely in the production of superoxides, notably for pathogen defense and ROS signaling [17].

A core set of ROS-related genes are conserved across all life as they evolved in early aerobic organisms before the split into modern kingdoms. Nonetheless, the ROS gene network has subsequently diversified across the different extant lineages. Prokaryotes presumably use an ROS maintenance system reminiscent of that employed by early cellular life on earth, though it has currently only been extensively studied in bacteria. Bacteria utilize the core ROS-scavenging systems of Fe/Mn-SODs, CAT and peroxidases/Prxrs, but often lack components found in more complex organisms, such as the nonenzymatic antioxidants GSH, AsA and tocopherols [18,19]. Unlike in Eukaryotes, cellular respiration is not believed to be a major source of cellular ROS in bacteria [20,21].

Eukaryotes share most of the ROS-related gene systems outlined in Table 1. However, there are some substantial or subtle differences in ROS metabolic systems across Eukaryotes or across specific lineages. Photosynthetic organisms, including some protists and most green algae and land plants, encode conserved genes related to the AsA–GSH cycle that are not present in nonphotosynthetic species (Figure 1B) [4]. The antioxidant AsA, rather than GSH, is thus more important in photosynthetic species, though AsA biosynthesis pathways differ between protists and green plants [22]. Animals primarily utilize the GSH redox cycle over AsA and in mammals the levels of AsA are believed to be too low to function as an effective superoxide scavenger except in specific tissues [10]. Primates and a few other species lack the ability to synthesize AsA entirely, instead relying on uptake of AsA from their diet. In contrast to other Eukaryotes, fungi do not produce AsA but have instead evolved to use AsA-analogs, notably D-erythroascorbate [10,23].

Several specific lineages also lack otherwise conserved ROS-related genes. The malaria parasite, *Plasmodium falciparum*, does not encode a true catalase or GPX gene and instead relies on the TRX redox cycle and import of host proteins to supplement this pathway [24]. These genes are also absent or nonfunctional in trypanosomatid parasitic protists, such as *Leishmania* [25]. In addition, Trypanosomatids produce the unusual GSH molecule trypanothione and express trypanothione-dependent reductase and peroxidase enzymes that function analogously to GR and GPX in trypanothione redox cycling [26,27]. Insects largely lack the GR and GPX genes and so, although they rely on the GSH redox cycle, GSH is regenerated via the TRX system rather than GR. Thus, differences in environment and lifestyle can result in drastic changes to the core Eukaryotic ROS network in certain lineages.

Extremophiles are organisms that thrive in the most extreme environments on Earth. Extremophile (and extremotolerant) organisms are common in prokaryotes, but can also be found throughout most Eukaryote kingdoms [28,29]. Fungi, as one of the most diverse Eukaryote lineages, contain many extremophile, or extremotolerant, species [30]. Several extremotolerant plant species display resistance to high salt, high UV, desiccation and heavy metals. In animals, classical extremotolerant traits are largely restricted to microscopic species. Dormant tardigrades in their tun (resistant) form can survive temperature ranges from near-absolute zero to 151 °C, pressure ranges from vacuum to 6000 atmospheres, total desiccation and exposure to ionizing radiation [31]. A small handful of vertebrate species are tolerant of freezing [32] and, though not typically referred to as an extremotolerant species, the naked mole rat *Heterocephalus glaber* displays many unique traits compared to other mammals due to its harsh subterranean environment [33].

Anhydrobiosis is a common form of extremotolerance observed in complex Eukaryotes. The origin of land plants is hypothesized to be linked to the evolution of desiccation tolerance (DT) in early aquatic algae [34]. Vegetative DT is still common in simple plants such as hornworts, clubmosses and mosses, and has re-evolved independently in angiosperms multiple times [35]. Several microscopic animals are DT, including tardigrades, nematodes and rotifers, though in complex animals only the larvae of the chironomid insect *Polypedilum vanderplanki* show DT [36,37]. The damaging effects of ROS are thought to be a major barrier to survival during anhydrobiosis [38], and the up-regulation of genes associated with maintenance of ROS homoeostasis is commonly reported in DT species [39]. Antioxidant status is a predictor of survival of the resurrection plant *Myrothamnus flabellifolia* [40], and poikilochlorophyllous resurrection plants in regions with high irradiance degrade chlorophyll during desiccation to prevent passive production of ROS [34,35]. Desiccation tolerant and sensitive tardigrade species show differences in the induction of ROS-related genes, and bdelloid rotifers are highly resistant to ionizing radiation-induced oxidative damage through a particularly effective antioxidant system [41,42]. Additionally, rotifers are the only animals that produce trypanothione, probably via HGT-derived metabolic pathways, and trypanothione genes are up-regulated during desiccation [43].

The aim of this study is to analyze the complement of ROS network genes and gene families across a range of species from all Eukaryotic kingdoms, with a special emphasis on extremotolerant species. These results provide insight into the effects of commonly encountered extreme environmental conditions on ROS-related gene family number and organization in tolerant versus sensitive species.

## 2. Results

### 2.1. Evolution of the ROS Network Genes across Eukaryotes

The aim of this study was to compare the ROS gene network across Eukaryotes and additionally between species tolerant and sensitive to extreme conditions (Table A1). Genome sequencing projects have been performed on a range of model Eukaryotic species and, over the past few years, genomes from extremotolerant species, particularly resurrection plants, have also become available [41,44,45,46,47,48]. However, relatively few genomes have been sequenced from fungi, protists and nonangiosperm plants (mosses, lycophytes and ferns) at this time [49,50]. In total, we analyzed the genomes of 37 plants (including eight extremophiles) and 28 nonplant species (including eight extremophiles). For the sake of this study, putative homologs are identified using the common names for the gene family (e.g., NOX, GPX). However, as these genes are identified by homology alone, there is no guarantee that they are functional or that the method of action is the same as those typically found in model organisms. In addition, the absence of an orthologue related to a chemical pathway in an organism does not necessarily imply that that organism is unable to catalyze that reaction—it could instead be achieved through species, or lineage-specific homologues or redundant mechanisms.

The proportion of single-gene and multigene OrthoFinder orthogroups (OGs) was investigated in each analyzed species (see Methods). Plants generally had the smallest proportion of single-gene OGs compared to both animals and less complex organisms (Figure A1; *p* = 1.714 × 10^−9^), likely reflecting the history of whole genome duplication (WGD) events in this lineage. The rotifer *Adineta vaga* showed a very divergent pattern of gene duplication compared to all other species in that it was enriched for two- and four-gene OGs (Figure A1). Rotifers are tetraploid with genes often occurring as four copies arranged in pairs, consistent with these results [48].

Our analysis indicates that ROS gene families tend to contain multiple genes (Figure 2). Many ROS genes appear to be core to all Eukaryotes, including SODs (Mn-SOD and Cu-SOD), typical CATs, GR, GPX and other Prxrs (TII-, 2-Cys and 1-Cys Prx) and ferritins. Genes from these families are found throughout the Eukaryote lineage and were likely present in ancestral Eukaryotes. However, despite their ubiquity throughout the sampled species, genes from these families can nonetheless be absent from specific lineages. Catalase is absent from the malarial parasite, *P. falciparum*, as well as several green algal species, for instance (Figure 2). *Drosophila melanogaster* as well as all other investigated winged insects encode no GR, despite its prevalence throughout other Eukaryotes (Figure 2). Photosynthetic organisms, and especially higher plants, show the largest accumulation of unique ROS-related genes and gene family expansions. This is largely due to the presence of additional photosynthesis-specific genes, divergence of additional ROS gene isozymes that have localized to plastids, plastid-derived genes (such as chloroplast Fe-SODs) and the history of lineage-specific genome duplications in land plants (Figure A1). Conversely, animals display a diversification of NOX-like genes compared to the plant RBOHs (NOX1-4, NOX5 and DUOX families), animal-specific PRX and putative Cu-only SOD-repeat proteins (Figure 2). However, when comparing the number of genes in each family across extremophile and sensitive species we find no clear evidence for consistently expanded ROS-related gene families in extremophiles (Figure 2).

The presence of ROS-related genes was determined based on the publicly available genomes of the selected Eukaryotic species, and thus the confidence of the data is dependent on the gene annotations for these species. Assembly errors, gaps, contamination or improper gene predictions can lead to annotation errors. Some of the analyzed species lack genes from multiple core ROS-related gene families, for example *Gnetum montanum* and *Xeromyces bisporus* (Figure 2). In these instances, it is likely that these genes are absent or unannotated in the currently assembled genome but not necessarily that these species truly lack so many core genes. Several species also contained ROS-related gene sequences that appeared to be only distantly related to other Eukaryotic genes in the family, at basal positions of the gene phylogenies. Often these could be identified as potential contamination based on near-perfect sequence similarity to a known bacterial sequence, and/or by the location of the gene on a small scaffold containing only genes of predicted bacterial origin. However, in some cases—such as in tardigrades and rotifers—frequent HGT events have resulted in the confident prior classification of horizontally derived genes. In the following subsections we discuss the findings for each of the gene families shown in Figure 2.

### 2.2. Superoxide Dismutases

#### 2.2.1. Fe/Mn-SOD

Fe-SODs were found predominantly in photosynthetic species and generally localized to the chloroplasts (Figure 3A). Angiosperms had a notable expansion of Fe-SODs early in the lineage compared to other investigated land plants, though we found none in the available assembly of the resurrection plant *Oropetium thomaeum*. Fe-SODs were also absent in *Klebsormidium nitens* and all Mamiellales green algae (*Micromonas commoda*, *Ostreococcus tauri* and *Bathycoccus prasinos*) in this study. The nonphotosynthetic protozoan parasite *P. falciparum* contains an Fe-SOD, but it is unclear how common they are in protists in general [14].

Mn-SODs were found in nearly all Eukaryote species and were predicted to be predominantly localized to the mitochondria (Figure 2 and Figure 3A). There is a clear separation between the Mn-SOD genes found in green plants and those in animals and fungi, whereas the phylogenetic distribution of protist, diatom and red/brown algal Mn-SOD genes is less clear-cut. The green algae *Chlamydomonas reinhardtii* and *Volvox carteri* and the nongreen algae *Cyanidioschyzon merolae*, *Ectocapus siliculosus* and *Phaedactylum tricornutum* contain an expanded repertoire of Mn-SOD genes. Unlike other Eukaryote Mn-SODs, these proteins are predicted to be extracellular or associated with the chloroplast or cell membranes, and a novel chloroplast-targeted Mn-SOD has previously been identified in *C. reinhardtii* [73]. As both *C. reinhardtii* and *V. carteri* (and *C. merolae*) lack Cu-SODs, these additional Mn-SODs may help to augment other Fe/Mn-SODs under oxidative stress. No Mn-SODs were identified in the dicot *Xerophyta viscosa*, the gymnosperms *Cycas micholitzii*, *Picea abies* and *G. montanum*, and the fungi *X. bisporus*, though it is unclear to what degree this may be due to incompletely assembled or unannotated genome sequences.

Several individual genes appear to be only distantly related to typical Eukaryote Fe/Mn-SODs, such as examples from *Ricinus communis*, *Pinus sylvestris* and *P. vanderplanki* (Figure 3A). The high similarity of these genes to bacterial proteins and the genomic context in which they are usually found (small, isolated scaffolds often containing other bacterial genes) suggests that these are likely present due to sample contamination. On the other hand, we find evidence for putative horizontal gene transfer in at least a few species. The leech *Helobdella robusta* contains an Mn-SOD that may be derived via HGT from a prokaryote source, and the rotifer *A. vaga* contains two pairs of such genes. In both cases the resultant Mn-SOD proteins are not predicted to be mitochondrial, but either cytoplasmic or extracellular/membrane-associated.

#### 2.2.2. Cu/Zn-SOD & Ni-SOD

Most Cu/Zn-SODs were found within a single OG that encompassed proteins from all Eukaryotic groups (Figure 3B). Plant Cu/Zn-SODs clustered separately from those of other Eukaryotes and were further divided into three clades: a chloroplastic form and two cytoplasmic forms. Chloroplastic Cu/Zn-SODs were found across nearly all green plants, whereas the cytoplasmic forms were restricted to multicellular plant species. Cu/Zn-SODs were similarly prevalent in animals, particularly invertebrates. They appeared to be divided into a cytoplasmic form (found in nematodes, insects and vertebrates, but also some fungi, protists and nongreen algae), and a predominantly extracellular form found across all animals. Protists generally do not contain Cu/Zn-SOD; however, multiple cytoplasmic Cu/Zn-SODs were identified in both *Dictyostelium discoideum* and *Tetrahymena thermophila* [14].

Further, two additional Cu/Zn-SOD OGs were identified: vertebrate-specific extracellular Cu/Zn-SODs were in a separate OG to other Cu/Zn-SODs, though the protein sequences clustered within the phylogeny of invertebrate extracellular Cu/Zn-SODs (Figure 3B). A third OG contained predominantly Cu-only SOD repeat proteins (CSRPs), previously described in several animal species [74]. Unlike typical Cu/Zn-SODs and the Cu-only SOD proteins found in some mycobacteria and fungi, CSRPs contain multiple tandem Cu-SOD-like domains [51,75]. In this analysis, CSRPs were found in both tardigrade species, all insects and aquatic animal species (oyster, sea sponge, lamprey and fish) (Figure 3B).

Ni-SODs are thought to have evolved after the split between prokaryotes and Eukaryotes and are found in only a few bacteria lineages [14]. However, genes containing an Ni-SOD domain fused to polyubiquitin were identified in all three Mamiellales green algae (*B. prasinos*, *M. commoda* and *O. tauri*), and the diatom *P. tricornutum*. It is unclear whether these putative Ni-SODs are functional, though it is interesting to note that the Mamiellales also appear to lack an Fe-SOD gene that is conserved in nearly all other green plants [76].

### 2.3. Catalases

Only typical catalases and catalase-peroxidases are found in Eukaryotes, and the latter have not been reported in animals or higher plants (Figure 4A). All identified typical catalases were found in a single OG and shared the PF00199 (catalase) and PF06628 (catalase-related) protein domains. As expected, typical catalase genes were identified in nearly all species across all Eukaryotic domains and were generally associated with the peroxisomes (Figure 4A). The exceptions to this, in our study, were the human malaria parasite *P. falciparum*, the gymnosperm *G. montanum*, and unicellular algae of the order Mamiellales (*M. commoda*, *O. tauri* and *B. prasinos*). *P. falciparum* has been reported to not encode a catalase and to instead rely on thioredoxins and host Prx enzymes to detoxify H_2_O_2_ [24]. The consistent absence of catalase across the analyzed Mamiellales algae also suggests that they may not be reliant on catalase for the removal of H_2_O_2_. Catalase protein domain structure was highly consistent across all Eukaryote catalases, with the exception of the tardigrades *Hypsibius dujardini* and *Ramazzattius varieornatus*. Catalases from both species contained an additional PF01965 (DJ-1/ThiJ-family) domain, typically associated with the clade II bacterial typical catalases [77], and are believed to have been acquired via HGT [41].

Catalase-peroxidases were found in a single OG and contained only the PF00141 (peroxidase) protein domain. Catalase peroxidases were identified in the genomes of the fungi *A. nidulans*, the Chlamydomonadales green algae *C. reinhardtii* and *V. carteri*, the diatom *P. tricornutum* and the brown alga *E. siliculosus*. Apart from *E. siliculosus*, which had eight genes, each species contained only a single catalase-peroxidase. A catalase-peroxidase gene was also identified in the genomes of the moss *Physcomitrella patens* and the dicot *Boea hygrometrica*. However, these proteins were highly similar to bacterial catalase-peroxidase sequences and were encoded on small genome scaffolds that contained other genes of probable bacterial origin, and thus most likely represent contaminant gene sequences.

There appear to have been expansions of the catalase or catalase-peroxidase gene families in the genomes of the fungi *A. nidulans*, the rotifer *A. vaga*, the insect *Bombyx mori*, the brown algae *E. siliculosus* and the moss *P. patens*. Such expansions have previously been reported for both *A. nidulans* [78] and *B. mori* [79]. However, there was no general correlation between the number of catalase genes and extremotolerance.

### 2.4. Ferritin

Ferritin is a ubiquitous iron-sequestering protein found in nearly all Eukaryote species (Figure 2). All identified ferritin genes were found in a single OG and shared the ferritin domain (PF00210). Many also contained a rubrerythrin domain (PF02915). Plant and animal ferritins clustered separately during phylogenetic analysis (Figure 4B). Animal ferritins were predicted to be either cytosolic or extracellular; in contrast, most plant ferritins were predicted to be targeted to the chloroplast (in angiosperms) or chloroplast and/or mitochondria (in algae, moss, ferns and gymnosperms). Apart from *X. bisporus* and *C. merolae*, ferritins were absent from the fungi, protists and red/brown algae analyzed in this study.

### 2.5. Heme Peroxidases

Only the animal PRX superfamily and the APX/POX nonanimal PRXs were identified in the analyzed genomes; fungal ligninases and DyP-type PRXs were absent from all species. APX-family PRXs were found across three OGs, corresponding to nonplant cytochrome c peroxidases (CCPs) and plant cytosolic/peroxisomal APX; chloroplastic APX; and APX6-like genes (Figure 5A). Animal PRXs (XPOs) were all found in a separate OG (Figure 5B).

CCPs are related to plant APX genes and several were identified in the genomes of nonplant species, though their relationship was not neatly resolved (Figure 5A). CCP and APX genes all shared the PF00141 (peroxidase) PFAM domain. It is unclear to what degree the presence of these genes is associated with an early Eukaryotic ancestor, HGT or contamination (for example, such genes found in the genome of *P. sylvestris* and *Beta vulgaris*).

On the other hand, plant APX proteins clearly clustered into four classes corresponding to the *Arabidopsis thaliana* APX6-like, chloroplastic (thylakoid and stroma), peroxisomal and cytosolic forms (Figure 5A). The chloroplastic form was found across all photosynthetic organisms—red/brown algae, green algae, and lower and higher plants. APX6-like APXs were similarly found throughout the plant kingdom but were specifically absent from gymnosperms. Peroxisomal and cytosolic APXs were found only in plants, except for a single *K. nitens* protein. CELLO was unable to detect peroxisomal targeting in the genes from this clade, though there is substantial evidence that these genes are targeted to peroxisomes.

The animal PRX superfamily shared the animal peroxidase (PF03098) protein domain (Figure 5B). Animal PRXs were restricted entirely to animal species except for several genes identified in the brown alga *E. siliculosus* and a single gene in the green alga *V. carteri*. Animal PRX genes were far more abundant in the genomes of invertebrates than vertebrates, and substantially expanded in rotifers, tardigrades and the Pacific oyster, *Crassostrea gigas*.

The remaining heme peroxidases—the classical plant PRXs (POX)—consisted of nearly 1500 genes found across 11 OGs, restricted almost entirely to plants. The function of most POX proteins is currently unclear, but the ubiquitous presence and large number of these genes suggests that they are important for cellular functioning.

### 2.6. Non-Heme Peroxidases (Prx & GPX)

Non-heme PRXs were found across five different OGs, corresponding directly to GPX and the four Prx subfamilies (2-Cys Prx, 1-Cys Prx, TII-Prx and PrxQ). Most GPX and all Prxs shared the PF00578 (alkyl hydroperoxide reductase and thiol-specific antioxidant) and PF08534 (redoxin) domains, with subtle differences in domain layout. Additionally, GPXs contained the PF00255 (GSH-peroxidase) domain, and the 1-Cys and 2-Cys Prxs contained a C-terminal PF10417 (1-cysPrx C-terminal) domain (Figure 6).

GPX was found in nearly all analyzed species, consistent with its central role in ROS homeostasis (Figure 2). Plant and nonplant GPX sequences clustered separately, with the odd exception of *Caenorhabditis elegans* GPX-1, GPX-2 and GPX-7 which clustered with a small clade of fern GPX proteins. Green algae GPX was consistently more similar to that found in non-photosynthetic organisms than to higher plant GPX, suggesting that the duplication and diversification of GPX in higher plants occurred later [80]. Invertebrates, particularly insects, contained comparatively fewer GPX genes compared to other organisms (Figure 2 and Figure 6). The majority of GPX proteins across all species were predicted to be cytosolic or secreted, apart from a plant clade that was predicted to be targeted to the chloroplast and/or mitochondria.

There were substantial differences in the number of genes and predicted localization of each class of Prx across different organisms (Figure 2). The 1-Cys, 2-Cys and TII-Prxs were identified in all Eukaryotic families. PrxQs, by contrast, were absent from all animals except for the bdelloid rotifer. TII-Prxs were generally present as only a single copy in animals but were expanded in land plants to include subfamilies specific to multiple organelles. On the other hand, 2-Cys Prxs were generally expanded in animals compared to other lineages, having both cytoplasmic and mitochondrial forms. For the most part Eukaryotic genomes contained few PrxQ and 1-Cys Prx genes, where only a single copy was present in most organisms. The size of these gene families may be constrained in most organisms compared to other Prx genes.

### 2.7. GSH/TRX Reductases

GR and high-MW TR proteins were found together within a single OG, highlighting the close relationship between the two gene families (Figure 7). All GR and high-MW TR genes contained a pyridine nucleotide–disulphide oxidoreductase protein domain (PF00070, PF07992 or PF13738) and a pyridine nucleotide–disulphide oxidoreductase dimerization domain (PF02852). GR was identified across all Eukaryotic kingdoms and most species contained at least one gene copy—except for insects, the leech *H. robusta* and the protist *T. thermophila* (Figure 1). Some algae and most plants contained two or more copies of GR, one predicted to be targeted to the chloroplast and another to the cytoplasm (Figure 7). High-MW TR was found predominantly in animals (invertebrates and vertebrates), protists and red algae but was absent from all higher plants.

Low-MW TRs were identified in a separate OG, indicative of their distant relationship to GR/high-MW TR. They shared most protein domains with high-MW TR but lacked the dimerization domain. In contrast to high-MW TRs, they were found in plants as well as most protists, fungi and algae but were entirely absent from animals. This could suggest that both high- and low-MW TRs (or an ancestral version of these genes) were present in early Eukaryotes, including the direct ancestors of protists and algae, but subsequently diverged or were respectively lost in the lineages giving rise to fungi, animals and higher plants.

### 2.8. DHAR

DHAR is a plant-specific member of the glutathione S-transferase (GST) gene family. Plant DHAR genes were found within a single OG, although they clustered together with several other related GST families within this OG. Only the clade consisting of DHARs is shown in Figure 8A. All DHARs contained the specific GST N- and C-terminal PFAM domain and, in addition, several also contained a glutaredoxin domain (PF00462).

DHAR protein genes were found in nearly all higher plant genomes (except for *G. montanum* and *C. micholitzii*) and in the green algae *C. reinhardtii* and *K. nitens*. Angiosperm DHARs are split into two subfamilies, corresponding to the proteins predicted to be targeted to the chloroplast/mitochondria and the cytoplasm (Figure 8A). A DHAR-like gene was also identified in the brown algae *E. siliculosus* and the diatom *P. tricornutum*. DHAR genes have previously been identified in red algae [81] though not in *C. merolae*, the only red algae analyzed in this study.

### 2.9. MDHAR

All MDHAR queries were found in a single OrthoFinder OG that contained not only MDHAR proteins but also the closely related apoptosis inducing factor, mitochondrial (AIFM) FAD-dependent oxidoreductases AIFM1 and AIFM3, albeit with clear division between the AIFM and MDHAR subfamilies. The MDHAR subfamily was additionally divided into the chloroplastic/mitochondrial, peroxisomal and cytoplasmic MDHAR clades (Figure 8B). However, CELLO was unable to detect peroxisomal targeting motifs in proteins from the known peroxisomal clade.

As would be expected, MDHAR genes are present only in the genomes of photosynthetic organisms—predominantly higher plants (Figure 2). Red and brown algae (*E. siliculosus* and *C. merolae*) make the base of the MDHAR phylogeny, and green algae generally contain only a single chloroplastic MDHAR gene. *K. nitens* is the only noncomplex plant to contain an MDHAR gene from all three clades. Most higher plants contain at least three MDHAR genes, often more (Figure 2 and Figure 8B). However, there was no evidence that extremotolerant plants contained an expanded MDHAR gene family compared to other species.

The moss *P. patens* has been reported to lack a chloroplastic MDHAR [58] and no such gene was identified in this study. The fern *Salvinia cucullata*, the gymnosperms *Ginko biloba* and *P. abies*, the basal monocot *Zea mays* and the grass *Zoysia japonica* also appear to lack chloroplastic MDHAR (Figure 8B).

### 2.10. AOX

The Eukaryote AOX gene family was found across two OGs. One OG contained protein sequences with similarity to *A. thaliana* chloroplastic *AOX4*, found specifically in algae and plants. The second OG contained genes similar to the mitochondrial forms of *A. thaliana* AOX proteins. All proteins contained the PF01786 (AOX) PFAM domain, and most also contained a COQ7 ubiquinone synthesis protein domain (PF03232), especially AOX proteins from higher plants (Figure 8C). Nearly all AOX proteins were predicted to be targeted to the mitochondria (Figure 8C).

AOX-coding genes were also identified in the genomes of the protist *T. thermophila*, the bdelloid rotifer, both tardigrade species and the Atlantic oyster, *C. gigas*. Though primarily associated with plants, there is increasing evidence that AOX can be found across some species from all Eukaryotic kingdoms [82].

### 2.11. NOX-like

The NOX-like family of genes were found in two OGs, corresponding respectively to the fungal FREs, and the plant FRO metalloreductases (branch collapsed) and ROS-producing NOX-like genes (Figure 9). Both FRE/FROs and NOX-like genes shared the NAD-binding (PF08030), FAD-binding (PF08022) and ferric reductase (PF01794) protein domains. Nearly all identified NOX family genes were predicted to be localized to the plasma membrane.

As would be expected, fungal FRE metalloreductases were restricted to fungi, though no FREs were identified in *X. bisporus* (Figure 2). The ROS-producing NOX genes were divided into distinct clades. The largest clade contained the plant-specific RBOHs, found specifically in the green algae *K. nitens* and higher plants. In contrast, the NOX1-4 (lacking an EF-hand domain), NOX5 and DUOX clades were found almost exclusively in animals (Figure 9).

Unicellular species contained few or no NOX genes, an observation in support of the association of NOX genes with multicellularity (Figure 2 and Figure 9). The slime mold *D. discoideum* goes through both a unicellular and multicellular life stage, which could explain the presence of animal NOX1-4/NOX5-like genes identified in this species. NOX-like genes have also been previously identified in several unicellular Eukaryotes, and given the family NOXD [17]. Several putative NOXD proteins were identified in the protist *T. thermophila*, the diatom *P. tricornutum*, brown alga *E. siliculosus* and the green algae *C. reinhardtii* and *K. nitens*. However, functional analysis is required to verify whether these genes are in fact ROS-producing oxidases.

## 3. Discussion

We analyzed the genomes of 65 diverse Eukaryotes to identify putative orthologs of core ROS-related genes (Table 1). The results of our comparative analysis are largely consistent with the understood mechanisms of ROS gene evolution in Eukaryotes as defined in previous literature [6,14,52,55,59]. In the context of ROS gene evolution this study not only adds data from a diverse array of Eukaryote species but also focuses on the little-explored question of how extreme stress tolerance may influence the evolution of these gene families in the genomes of extremophile Eukaryote species.

As would be expected, ROS-related gene families generally consisted of multiple genes in each organism. Plants specifically show a tendency to expand the size of ROS-related gene families compared to other species. For example, many angiosperm genomes contained two or more copies of Mn-SOD, catalase or GR, whereas most animals, particularly vertebrates, encoded only a single copy (Figure 2). To a degree this may be due to the general expansion of all gene families in plants (Figure A1), but it is likely primarily related to the additional oxidative burden borne by plants due to photosynthesis. Additionally, most photosynthetic organisms encoded multiple plastid- or organelle-specific variants of key ROS genes, which are obviously absent from nonphotosynthetic organisms.

Several gene families, particularly SODs (both Cu/Zn and Fe/Mn groups), catalases and components of the GSH redox cycle (GR, GPX and Prxrs), were found across all analyzed Eukaryotic clades in this study (Figure 2). These genes are hypothesized to have been present in the common ancestor of all Eukaryotes, and generally serve an irreplaceable function: for example, detoxification of superoxides (SODs) and H_2_O_2_ (catalase). In contrast, several gene families are restricted to broad Eukaryote classes. The most obvious of these are genes involved in the AsA-GSH cycle (APX, DHAR and MDHAR), which evolved in early photosynthetic Eukaryotes and are found only plants and algae (Figure 2). Interestingly, DHAR was restricted to the genomes of complex plants with the exception of the green algae *C. rheinhardtii* and *K. nitens*. NADPH oxidases (NOX and RBOH in plants), which are important for intercellular ROS signaling, are assumed to be important for—and thus restricted to—multicellular organisms. Our results are generally consistent with this hypothesis, as the main NOX/RBOH classes are absent from single-cellular species analyzed in this study, including most protists and algae (Figure 2). The known exception to this is the slime mold *D. discoideum*, a model organism for research into the origins of multicellularity as it switches between a single-cellular and multi-cellular life cycle, and which contains several NOX genes similar to those found in animals (Figure 2). However, we did identify several novel NOX-like genes in several species that lacked typical NOX/RBOH genes: the unicellular protist *T. thermophila*, red algae *C. merolae*, and green algae *C. reinhardtii* and *O. tauri*, as well as the filamentous brown alga *E. siliculosus* and green alga *K. nitens* (Figure 2). These genes showed similarity to a class of NOX-like genes previously identified in red algae (including *C. merolae*), and named “NoxD” by the authors [17]. Our results suggest that this class of NOX-like genes may be found in red, brown and green algae, and does not appear to be restricted by multicellularity. However, it is still unclear whether these NOX-like genes are functional NADPH oxidases or a distinct class of metallo/oxidoreductases [17].

It is also known that some Eukaryotic lineages lack specific components of the ROS gene network or encode unique genes. *Drosophila melanogaster* lacks a GR gene [83], and regeneration of GSH is achieved through the TRX redox cycle. Our results indicate that this is likely also true for the honey bee, *A. mellifera*, the silkworm *B. mori* and both species of analyzed chironomids, *P. vanderplanki* and *P. nubifer* (Figure 2). This is particularly interesting in the case of *P. vanderplanki*, which produce desiccation-tolerant larva, where presumably the TRX cycle (or other redundant antioxidant systems) can maintain ROS homeostasis even in the absence of GR during desiccation stress. We also observed a lack of GR in another invertebrate, the leech *H. robusta*, as well as the fungi *X. bisporus* and the protist *T. thermophilus*. However, it is unknown whether these species, too, use TRX-based mechanisms to regenerate GSH. CSRPs (copper-only SOD-repeat SOD) are a newly defined class of putative SOD proteins. Unlike Cu-only SODs, which are found in some bacteria and fungi, or the typical Fe/Mn-SOD and Cu/Zn-SODs found in all Eukaryotes, CSRPs have so far been identified only in animals, specifically only in aquatic species and winged insects [74]. Our results are consistent with this finding, where these genes were largely restricted to all winged insects as well as fish, Atlantic oyster, sea sponge and both species of tardigrade (Figure 2). Amongst aquatic animals, however, CRSPs were absent from both the rotifer and leech. Although CRSPs are found in both the sensitive and extremophile chironomid and tardigrade species, it would be interesting to analyze their role – if any – during desiccation in these species.

Despite the comparatively harsh lifestyles of most extremotolerant species, we found no general tendency for expansion of gene families related to ROS production or scavenging (Figure 2). The rotifer, *A. vaga*, did have noticeable expansions of several ROS-related gene families compared to other invertebrates, with more than twice as many genes in each family on average. However, as recent tetraploids, the rotifer generally contains more genes per OG than any other of the investigated species (Figure A1). Thus, it is likely that this expansion is not related to extremotolerance specifically. In some cases, mostly amongst animals, certain extremophiles did show extreme gene family expansions compared to other species. For example, the mole rat, *H. glaber*, encoded by far the largest number of ferritin genes of all analyzed species. However, the only other rodent, *M. musculus*, contained the second largest number of predicted ferritin genes (Figure 2). Both tardigrade species were enriched for Cu/Zn-SODs and 2-Cys-Prxs compared to all other species, including other invertebrates; however, as this expansion occurred in both the resistant and sensitive species, it is likely that this is a lineage-related observation (Figure 2). Similarly, although both *Lindernia* species encoded the highest number of Cu/Zn-SOD, GPX-like and MDHAR genes amongst higher plants, there were no substantive differences between the sensitive and tolerant species, suggesting that this is likely a trait of this plant clade (Figure 2).

Most extremophiles display high constitutive expression of ROS scavenging systems or up-regulation of ROS-related genes during harsh conditions, and in many cases antioxidant potential is a good predictor of survival of extreme stress. The lack of expansion of ROS gene families in these species suggests that, in most cases, the basal ROS scavenging systems are sufficient to maintain ROS homeostasis even under the most extreme conditions. More insight into extremophile redox homeostasis might rather be gained by analyzing how, where and to what level the ROS genes are expressed in these species and how they jointly act in the context of the ROS gene network. Transcriptomics studies of sensitive and tolerant species subjected to varying types and levels of oxidative stress would provide a good basis for such an analysis.

## 4. Materials and Methods

### 4.1. Genomic Data

Genome data for the species used in the study were obtained from several sources, notably PLAZA (for most plant and red/green algae, https://bioinformatics.psb.ugent.be/plaza/ [84]), NCBI RefSeq (https://www.ncbi.nlm.nih.gov/refseq/ [85]), UniProt (https://www.uniprot.org/ [86]) and Ensembl (ftp://ftp.ensembl.org/ [87]). Other sources included FernBase (ftp://ftp.fernbase.org/ [88], http://download.tardigrades.org (tardigrade genomes) [41], http://bertone.nises-f.affrc.go.jp/(midge genomes) [89], http://www.genoscope.cns.fr/adineta/(Bdelloid rotifer genome) and http://thellungiella.org/ (*Eutrema salsugineum* genome) [90]. Genomes were also retrieved from the authors directly (*O. thomaeum*, *Lindernia brevidens* and *Lindernia subrecamosa*) [45,47] and *X. viscosa* [44].

### 4.2. Orthogroup (OG) Analysis

Detection of putative ortholog of ROS network genes across multiple species was performed using a combination of sequence homology to identify putative homologs, and orthology inference using OrthoFinder to refine a final list of putative orthologs sharing a common evolutionary origin. Putative homologs were identified based on sequence similarity using DIAMOND [91]. An initial set of query genes for each gene family over a range of species was obtained from UniProt. The list of all query proteins for each family was aligned to the proteomes of all species used in this study in an iterative fashion until no new hits were found. BLAST bit score ratio value (SRV) was used to threshold significant hits using SRV > 0.3 [92,93]. Separately, OrthoFinder was used to infer the evolutionary relationship between the proteins from all species and divide them into OGs composed of protein co-orthologues derived from the same ancestral gene [94].

Due to the diverse number of Eukaryote species analyzed in this study it would be expected that both phylogenetically and functionally related ROS genes could be found across multiple OGs if they shared a distant enough common ancestor, as might be the case with some lineage-specific genes. To get around this issue, the list of all putative homologs (which contains all publicly available protein data from the ROS gene families as well as homologs from this data set) was used to group OGs that contained functionally related ROS genes. Each OG was then interrogated—based on sequence similarity and the existence of the necessary PFAM domains—to verify that it contained putative genes from the ROS gene family of interest, rather than genes from a structurally similar but functionally unrelated family. OGs found to contain unrelated genes, as well as any protein sequences that were fragmented, chimeric or lacked relevant PFAM domains, were discarded from the analysis. This combinatorial approach allowed for the detection of distant homologues that would not have necessarily been identified based directly on their similarity to the proteins found in model organisms - and thus may lie in their own unique OG—while also excluding protein matches that are not evolutionarily related to the gene family of interest as determined by OrthoFinder. Presence of PFAM protein domains was calculated using hmmscan at an e-value cutoff of 1 × 10^−2^ and alignment coverage of 0.6 [95]. For each predicted orthologue the predicted subcellular localization was identified using CELLO v2.5 [96] using a database of Eukaryotic signal peptide sequences. Maximum-likelihood phylogenetic trees were calculated using FastTree [97] with default settings, and visualized together with PFAM and CELLO data using EMBL Interactive Tree of Life (https://itol.embl.de/) [98].

## 5. Conclusion

We performed a comparative genomics analysis, focusing on ROS network genes, across 65 diverse Eukaryotic genomes and including 16 extremophile species. Our investigation of the core ROS-related gene families (outlined in Table 1) is consistent with other more focused studies involving specific genes or species, as well as with conventional understanding of ROS gene evolution in Eukaryotes. However, despite analyzing genomes from all currently available extremophile species, we find no evidence for a general expansion of any ROS-related gene families in these species. We conclude that the basal ROS scavenging systems—found in extremophile and sensitive species alike—are sufficient to protect extremophile organisms even under the most adverse conditions.

## Figures and Tables

**Figure 1 ijms-21-09131-f001:**
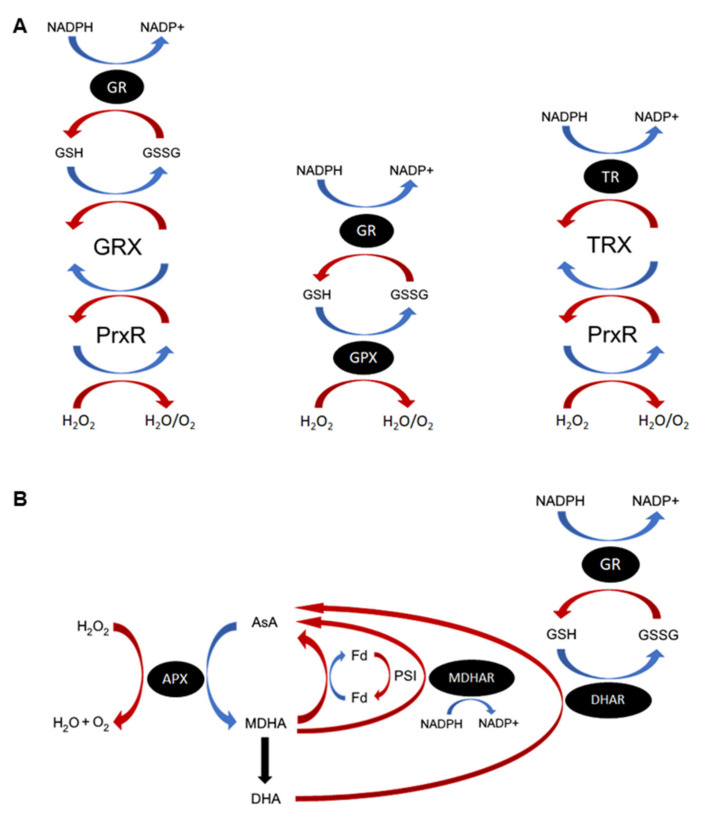
Important reactive oxygen species (ROS) antioxidant enzymes and cycles. (**A**) Functions of the glutathione (GSH), glutathione peroxidase (GPX), thioredoxins (TRX), and glutaredoxins (GRX) proteins in ROS homeostasis. GPXs and peroxiredoxins (Prxs) detoxify cellular H_2_O_2_ using different electron donor substrates. GPX causes oxidation of GSH to GSSG directly, resulting in production of H_2_O and O_2_. Reduced GSH is regenerated through the action of glutathione reductase (GR). Prxs are oxidized by H_2_O_2_, producing H_2_O and O_2_. Depending on the Prx, it then reduces either GRX or TRX. Oxidized GRX is nonenzymatically restored by GSH, which is oxidized to GSSG and in turn regenerated by GR using NADPH. Thioredoxin is regenerated by the related TRX-specific reductase (TR). In plant chloroplasts, TRX is also reduced by a ferredoxin-dependent TR (FTR). (**B**) The plant AsA–GSH cycle. H_2_O_2_ is reduced to water and oxygen through the action of ascorbate peroxidase (APX) and oxidation of AsA, producing the monodehydroascorbate (MDHA) radical. MDHA either spontaneously degrades, producing dehydroascorbate (DHA), or is converted back to ascorbate by MDHA reductase (MDHAR) and NADPH. In the chloroplasts, reduction of MDHA occurs primarily through PSI-photoreduced ferredoxin (reFd). DHA is converted to AsA by DHA reductase (DHAR) using GSH as an electron donor, producing oxidized GSH (GSSG). Reduced GSH is replenished by glutathione reductase (GR). In nonphotosynthetic organisms, AsA is replenished directly by GRXs and other antioxidants in the GSH–GR cycle. Oxidation reactions are coloured blue, whereas reduction reactions are red.

**Figure 2 ijms-21-09131-f002:**
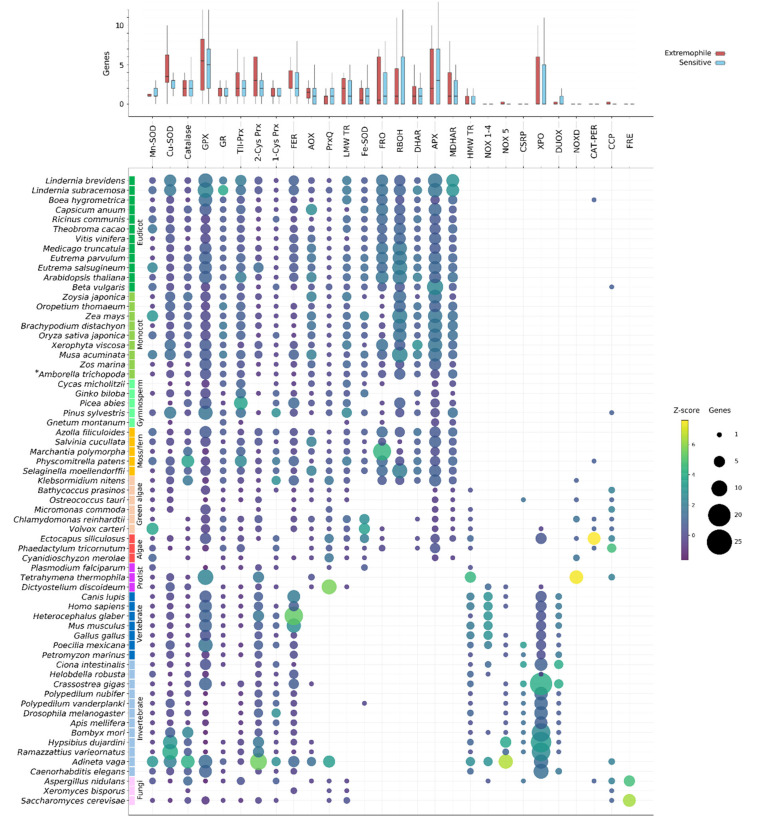
Presence and number of ROS network genes identified in the species analysed in this study. Upper panel: average number of genes in each family found in all extremophile and sensitive species (red and blue, respectively). No gene families were significantly expanded in extremophiles compared to sensitive species. Lower panel: The number of genes from each of the major ROS gene families identified in each species is shown by circle size, where circle colour corresponds to how closely that number compares to the mean number of genes in that family (Z-score). **Amborella trichopoda* is shaded as a monocot in this and other plots, though *Amborella* is its own clade of basal angiosperms.

**Figure 3 ijms-21-09131-f003:**
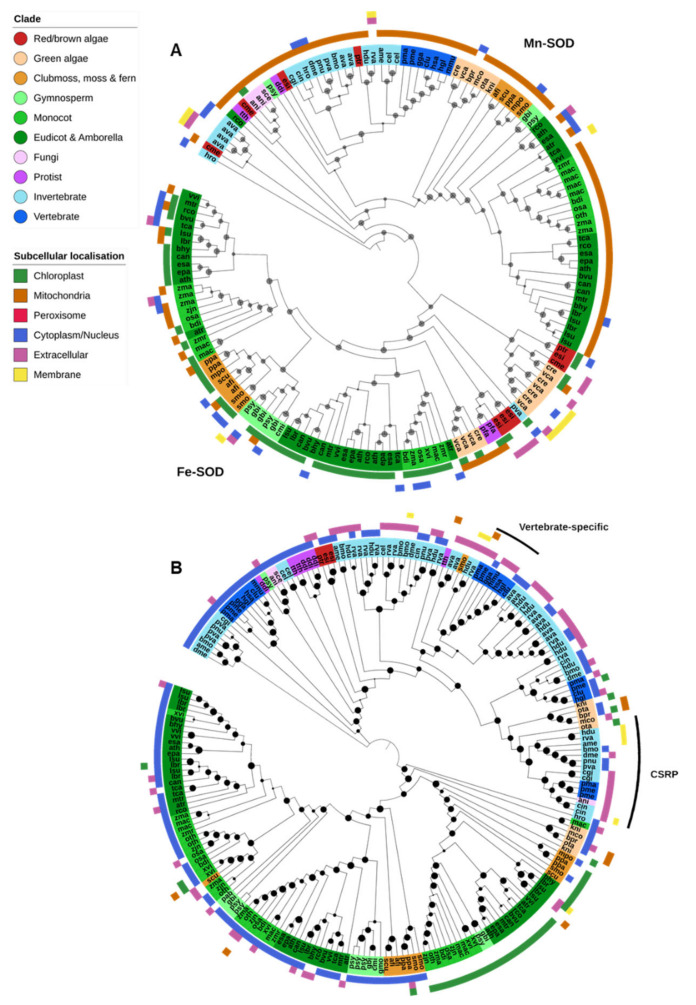
Superoxide dismutase. (**A**) Both Mn- and Fe-SODs were found in a single OG. Mn-SODs are found across all kingdoms, primarily in mitochondria, while Fe-SODs are found predominantly in photosynthetic organisms and targeted to the chloroplast. (**B**) Cu/Zn-SODs were found across all lineages. Vertebrate-specific extra-cellular Cu/Zn-SODs were found in a separate OG (marked with a black bar) though they clustered with other Eukaryotic Cu/Zn-SODs. A third OG contained Cu-only SOD repeat proteins (CSRPs). Maximum-likelihood phylogenetic trees were calculated using FastTree with default settings, and visualized together with PFAM and CELLO data using EMBL Interactive Tree of Life (https://itol.embl.de/).

**Figure 4 ijms-21-09131-f004:**
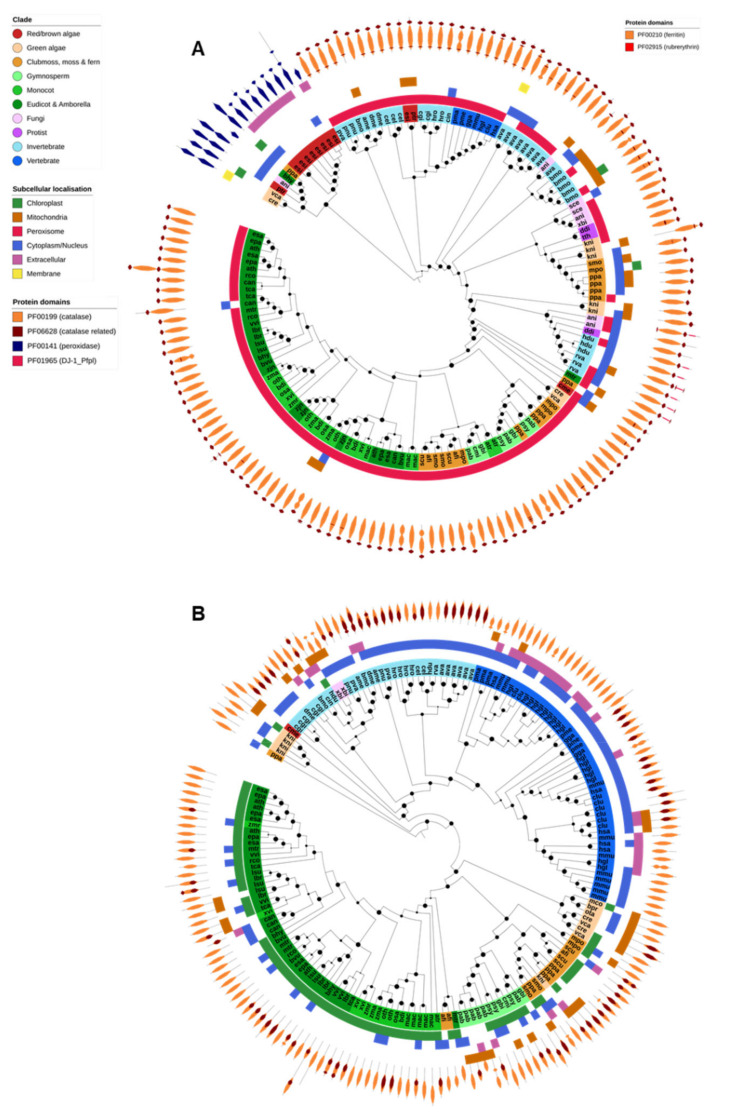
Catalase, catalase-peroxidase and ferritin. (**A**) The PFAM domain structure of each catalase gene is displayed on the outer ring of the tree, displaying the difference in domains between typical catalases (orange, primarily found in the peroxisomes) and catalase-peroxidases (blue, primarily secreted or found in the cytosol). All tardigrade catalase genes contain an additional PF01965 domain, common to bacterial catalases and likely evidence that they are derived via HGT. (**B**) Ferritins were found in nearly all species analysed and the gene family was expanded in some vertebrates compared to other species. In most organisms FER proteins were predicted to be cytosolic (blue outer band), with the exception of plants where they are found in chloroplasts (green outer band). The PF00210 (ferritin) protein domain was common to all proteins and the PF02915 (rubrerythrin) domain was additionally found across several organisms. Maximum-likelihood phylogenetic trees were calculated using FastTree with default settings, and visualized together with PFAM and CELLO data using EMBL Interactive Tree of Life (https://itol.embl.de/).

**Figure 5 ijms-21-09131-f005:**
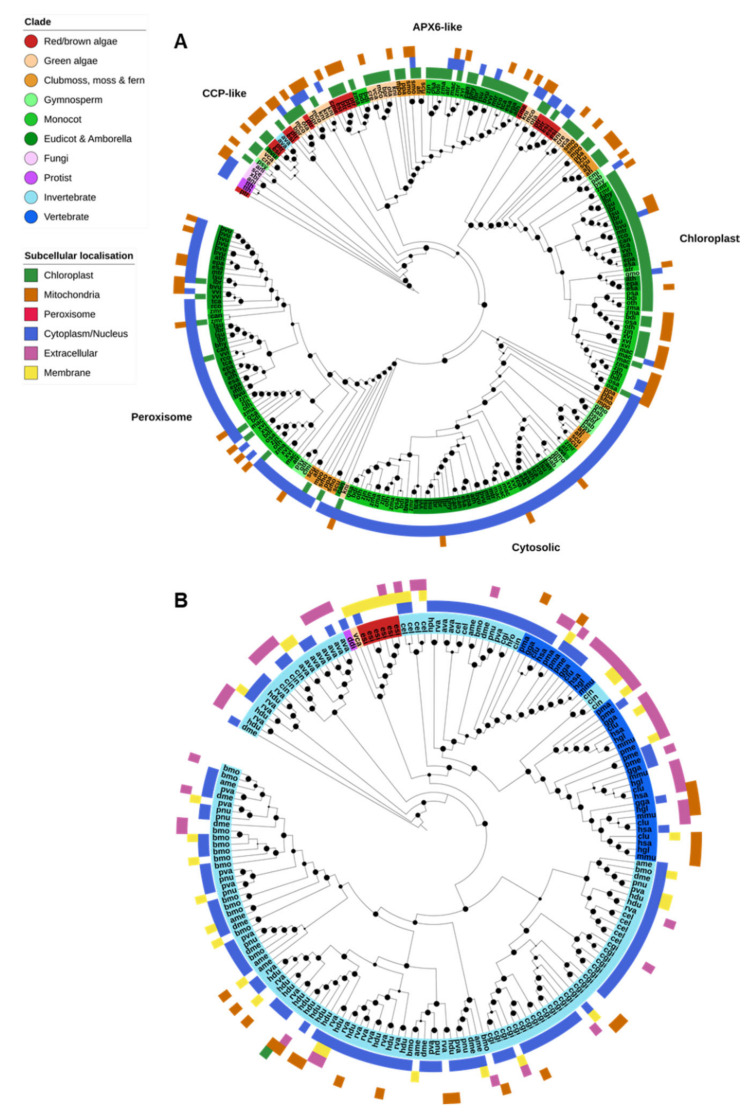
Heme peroxidases. (**A**) Plant ascorbate peroxidases (APX) were divided into four clades, corresponding to the APX6-like, chloroplastic, cytosolic and peroxisomal subfamilies. Cytochrome C peroxidases (CCPs) are related to the plant (APX) proteins and were found in the same orthogroup as plant cytosolic/peroxisomal APXs, though they formed a loose clade of largely algal, fungal and protistan proteins. (**B**) Animal-specific peroxidases (XPOs) were restricted almost entirely to animals. XPOs were massively expanded in invertebrates compared to vertebrates, though vertebrates contained their own clade of extracellular XPOs not found in invertebrates. Maximum-likelihood phylogenetic trees were calculated using FastTree with default settings, and visualized together with PFAM and CELLO data using EMBL Interactive Tree of Life (https://itol.embl.de/).

**Figure 6 ijms-21-09131-f006:**
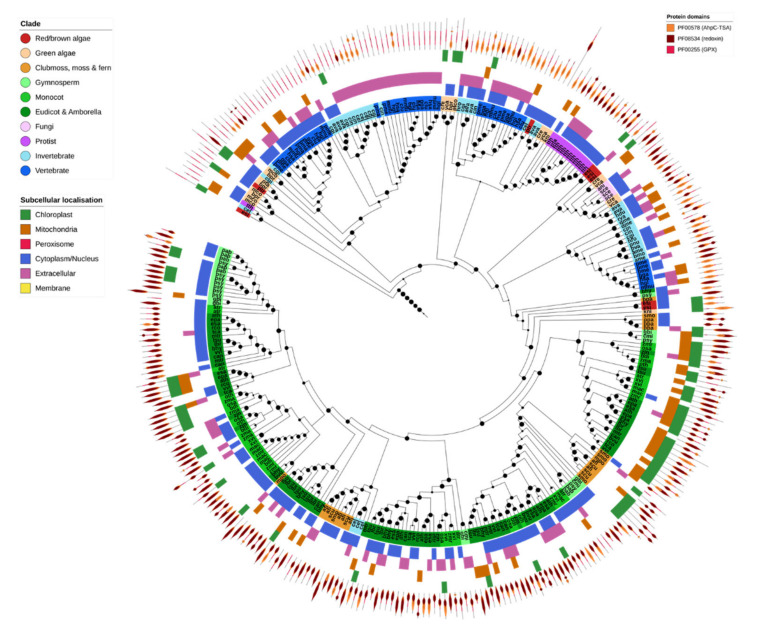
Glutathione peroxidases. Glutathione peroxidases (GPX) were common to nearly all species. Those from complex plants clustered separately to both green algae and other nonplant species. The protein structure for each gene is given in the outer ring of the tree, with the PFAM domain IDs given in the legend. Maximum-likelihood phylogenetic trees were calculated using FastTree with default settings, and visualized together with PFAM and CELLO data using EMBL Interactive Tree of Life (https://itol.embl.de/).

**Figure 7 ijms-21-09131-f007:**
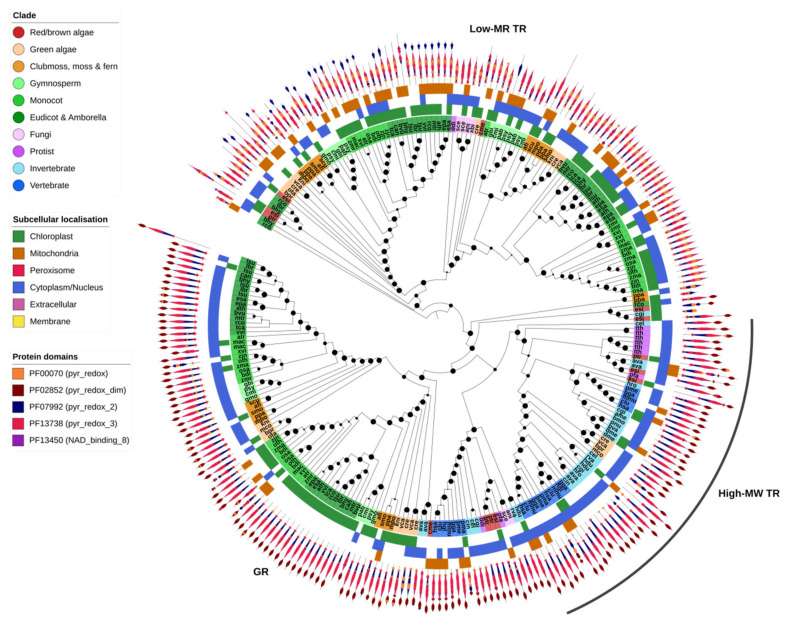
Glutathione/thioredoxin reductases (GR/TR). GR and high-molecular weight TR are closely related and were found in the same OG. GR was found across nearly all organisms, whereas high-MW TR was not found in complex plants. Instead, complex plants (as well as green algae and fungi) employ low-MW TR which is more distantly related to GR/high-MW TR. GR and both types of TR shared most protein domains, though low-MW TR lacked the PF02852 domain found in the other two protein families. Maximum-likelihood phylogenetic trees were calculated using FastTree with default settings, and visualized together with PFAM and CELLO data using EMBL Interactive Tree of Life (https://itol.embl.de/).

**Figure 8 ijms-21-09131-f008:**
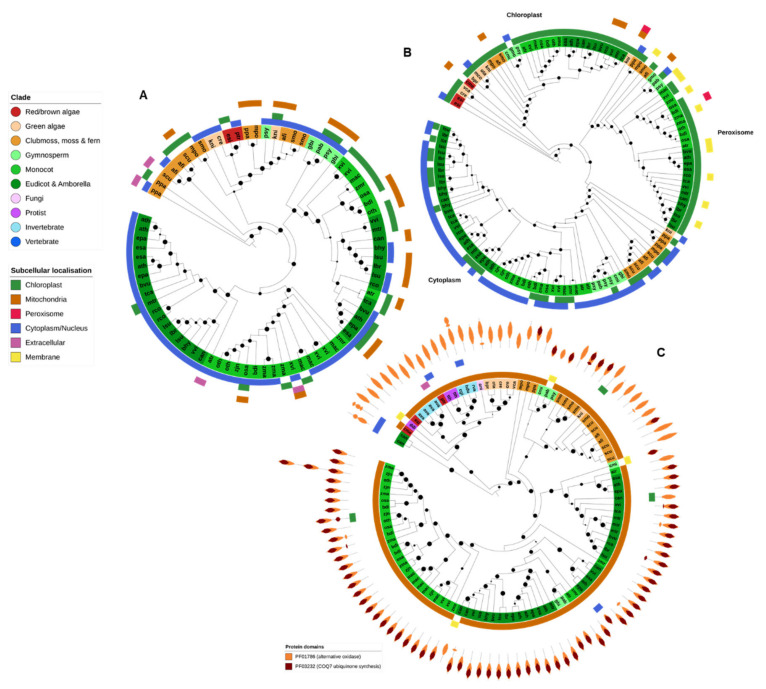
Dehydroascorbate reductase (DHAR), monodehydroascorbate reductase (MDHAR) and alternative oxidase (AOX). Dehydroascorbate reductase (DHAR) (**A**), together with monodehydroascorbate reductase (MDHAR) (**B**) are involved in ascorbate recycling during photosynthesis. As such, both gene families were found predominantly in photosynthetic organisms. DHAR from gymnospermms, eudicots and monocots were found in a separate clade to algae, moss and ferns, and further divided into chloroplastic and cytoplasmic forms. Complex plants generally contained at least three MDHAR variants (chloroplastic, peroximsonal and cytoplasmic) whereas algae contained only one. CELLO was unable to identify peroxisomal targeting motifs for the peroxisomal clade, which were instead predicted to be chloroplast and/or membrane. (**C**) AOX genes were found predominantly in algae and plants, with some additional examples in some stramenopiles, invertebrates and fungi. Nearly all AOX proteins were predicted to be mitochondrial. Maximum-likelihood phylogenetic trees were calculated using FastTree with default settings, and visualized together with PFAM and CELLO data using EMBL Interactive Tree of Life (https://itol.embl.de/).

**Figure 9 ijms-21-09131-f009:**
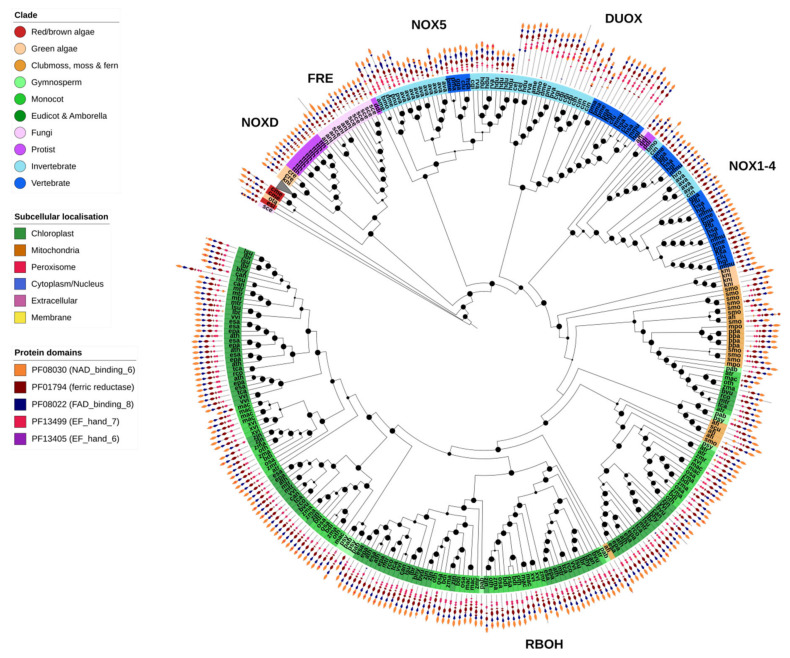
NADPH oxidase. NADPH oxidase (NOX) genes are involved in the production of ROS for the purposes of intercellular signaling. As such, their presence is largely restricted to multicellular organisms. Nonetheless, several NOX-like genes were identified in single-celled organisms in this study. The clade of plant ferric reductase oxidases (FRO) genes, which were found within the same OG as the plant NOX (RBOH) genes, are collapsed in this figure (grey triangle). Fungal FRE proteins are highly similar to FRO/NOX genes, but it is unclear how many FREs function as ROS signaling proteins as opposed to metalloreductases. Animals show several families of NOX genes with differing protein domain organizations compared to plant RBOHs (shown on the outer ring of the chart, legend on the left). Maximum-likelihood phylogenetic trees were calculated using FastTree with default settings, and visualized together with PFAM and CELLO data using EMBL Interactive Tree of Life (https://itol.embl.de/).

**Table 1 ijms-21-09131-t001:** Central enzymes involved in ROS homeostasis across Eukaryotes. Overview of the central gene families and sub-families functioning in ROS homeostasis in Eukaryotes, together with a summary of their mechanism and selected references.

Family	Sub-Family	Function and References
Superoxide dismutase (SOD)	Iron/Manganese SOD (Fe/Mn-SOD)	Catalyzes the detoxification of superoxide radicals into oxygen and hydrogen peroxide (H_2_O_2_). [14,51]
	Copper/Zinc SOD (Cu/Zn-SOD)	
	Copper-only SOD (Cu-SOD)	
	Copper-only SOD-repeat protein (CSRP)	
	Nickel SOD (Ni-SOD)	
Catalase (CAT)	Typical catalase	Catalyzes the dismutation of H_2_O_2_ into oxygen and water. [52,53]
	Catalase-peroxidase	
	Manganese-containing catalase	
Heme peroxidases (PRX)	Animal heme-PRX	Functions in the detoxification of H_2_O_2_ or peroxide radicals via the oxidation of a wide variety of organic and inorganic substrates, using either a heme or cysteine/selenocysteine cofactor. [54,55]
	Ascorbate PRX (APX)	
	Ligninase (LiP/MnP/VP)	
	Classical plant PRX (POX)	
	DyP-type	
Non-heme peroxidases (GPX, peroxiredoxins/Prx)	Glutathione PRX (GPX)	
	Type-II Prx	
	PrxQ	
	1-Cys Prx	
	2-Cys Prx	
Redoxins	Thioredoxin (TRX)	Involved in dithioldisulphide exchange, such as reduction of thiol-containing Prxs. [9]
	Glutaredoxin (GRX)	
Glutathione reductase (GR)	GR	NADPH-dependent maintenance of the cellular pool of reduced glutathione (GSH). [24]
Thioredoxin reductase (TR)	High-MW TR	NADPH-dependent maintenance of the cellular pool of reduced redoxins. [56,57]
	Low-MW TR	
Monodehydroascorbate reductase	MDHAR	Regenerates ascorbate (AsA) from MDHA radicals generated during the AsA-GSH cycle. [58]
NADPH oxidase (NOX)	NOX1-4	Membrane-bound oxidases that specifically produce superoxide. [5,17,59,60]
	NOX5	
	DUOX	
	RBOH	
	NOXD	
Alternative oxidase	AOX	Terminal oxidase in an alternative mitochondrial respiration pathway. [61,62]
Ferritin	Ferritin	Fe-sequestering proteins that play an essential role in iron homoeostasis. [63,64,65]
Glutathione S-transferase (GST)	Alpha	Catalyzes the conjugation of GSH to a multitude of activated xenobiotic substrates, including toxins and secondary metabolites. [66,67,68,69,70,71,72]
	Delta	
	DHAR	
	Ef1Bg	
	Epsilon	
	Hemerythrin	
	Iota	
	Lambda	
	Mu	
	Omega	
	Phi	
	Pi	
	Sigma	
	Tau	
	THCQD	
	Theta	
	Zeta	

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
