# Peer review of "Comparative Analysis of ROS Network Genes in Extremophile Eukaryotes"

_ijms, 2020, doi:10.3390/ijms21239131_

Round 1

Reviewer 1 Report

Lyall et al. conducted comparative genomics across diverse eukaryotic genomes to search for changes in reactive oxygen species pathway related to extremophile lifestyles. Although they observed a subset of ROS pathways that are specific to plants, they observed no connection between the copy number of ROS pathways and extremophile lifestyle. This suggests that although the ability to detoxify ROS is important for extremophiles, expansion in copy number or innovation of these genes is not broadly associated with this trait. This is not a surprising finding, but it represents a set of high-quality analyses that will be of great use for the extremophile community. The technical aspects of this manuscript are sound, and the bioinformatics analyses are robust. I have a few suggestions on how the results are presented that would strengthen the manuscript.  

  1. The central results of this paper are not surprising or particularly exciting, but they are important and should be discussed more in the main results. Supplemental Figure 2 provides a nice summary of the number of genes in each ROS-related gene family for extremophile and sensitive species. This figure shows that in general, there are not stark differences in ROS networks between tolerant and sensitive species, and tolerance is likely driven by other factors. I would suggest including Supplemental Figure 2 to the main text and including a section in the results discussing the summarized findings from each gene family. It is difficult for a non-expert to read through each section on the different ROS detoxifiers and draw meaningful conclusions.

  1. The title, abstract, and discussion focus on the finding that plants have sets of specific ROS network genes compared to other Eukaryotes. This is well-known and focusing on this aspect takes away from the central finding that the basal ROS scavenging systems are sufficient to protect eukaryotes in extreme conditions. I would suggest changing the title and abstract to highlight this interesting finding. Although a lack of expansion may be a negative result, it nonetheless advances our insights into the evolution of extremophiles.

  1. In Figure 1, the placement of several species is wrong. For instance, Ostreococcus is listed as a eudicot, Amborella is listed as a monocot, Oropetium as a green algae, etc. I am not sure if this is because the names are incorrect or the ordering was wrong.

  1. The details on how ROS pathway genes were identified across Eukaryotes is unclear. Many ROS related genes with the same function would cluster in different orthogroups across the diverse Eukaryotes and although the authors state that pfam domains were also used to identify ROS related genes, the pipeline of steps that were taken to assign genes to specific ROS pathways is unclear. Perhaps a section in the results describing the results from OrthoFinder (i.e. total number of orthogroups, number specific to plants, fungi, animals, etc) and any enrichments in particular groups would be interesting.

Author Response

  1. The central results of this paper are not surprising or particularly exciting, but they are important and should be discussed more in the main results. Supplemental Figure 2 provides a nice summary of the number of genes in each ROS-related gene family for extremophile and sensitive species. This figure shows that in general, there are not stark differences in ROS networks between tolerant and sensitive species, and tolerance is likely driven by other factors. I would suggest including Supplemental Figure 2 to the main text and including a section in the results discussing the summarized findings from each gene family. It is difficult for a non-expert to read through each section on the different ROS detoxifiers and draw meaningful conclusions.

Supplemental Figure 2 has now been added to Figure 2 (Figure 1 in the previous version of the manuscript) and so is now part of the main text. The Results/Discussion has been expanded to focus more on extremotolerance-related results in context.

  1. The title, abstract, and discussion focus on the finding that plants have sets of specific ROS network genes compared to other Eukaryotes. This is well-known and focusing on this aspect takes away from the central finding that the basal ROS scavenging systems are sufficient to protect eukaryotes in extreme conditions. I would suggest changing the title and abstract to highlight this interesting finding. Although a lack of expansion may be a negative result, it nonetheless advances our insights into the evolution of extremophiles.

The Title, Abstract and Discussion have been reformatted to focus more on the major finding (the lack of expansion of ROS-related gene families) and less on plants.

  1. In Figure 1, the placement of several species is wrong. For instance, Ostreococcus is listed as a eudicot, Amborella is listed as a monocot, Oropetium as a green algae, etc. I am not sure if this is because the names are incorrect or the ordering was wrong.

The names for Ostreococcus (otr) and Oropetium (oth) were swapped in the table, but the underlying data is correct. For the sake of this study Amborella has been coloured as a monocot to reduce the range of groups (and thus colours) required. Similarly, the green algae/diatoms, red/brown algae and protist clades could also be more granular but have been given only two groupings. This has been clarified in the Figure Legend.

  1. The details on how ROS pathway genes were identified across Eukaryotes is unclear. Many ROS related genes with the same function would cluster in different orthogroups across the diverse Eukaryotes and although the authors state that pfam domains were also used to identify ROS related genes, the pipeline of steps that were taken to assign genes to specific ROS pathways is unclear. Perhaps a section in the results describing the results from OrthoFinder (i.e. total number of orthogroups, number specific to plants, fungi, animals, etc) and any enrichments in particular groups would be interesting.

The highlighted problem (distinguishing structurally-similar, but phylogenetically- or functionally unrelated genes from ROS-related gene families) was the reason for using the approach outlined in the study. Lineage-specific ROS-related genes (for example, Cu/Zn-SODs specific to certain plants or animals) often formed an orthogroup separate from the more conserved genes in that family (the Cu/Zn-SODs found across all Eukaryotes). In this example., by merging all orthogroups that contained a canonical Cu/Zn-SOD from any annotated species, we were able to identify all such orthogroups. More rarely, related but functionally-divergent genes were placed in the same orthogroup (such as MDHARs and AIFM1/AIFM3 in plants). In this case, the phylogenic structure and occurrence/absence of specific PFAM domains could be used to separate such groups of genes. The pipeline used to identify ROS-related orthogroups, by merging putative homologs (determined by sequence similarity using BLAST) and phylogenetic evidence (using OrthoFinder) has been explained in more detail. In the same vein, this means that the raw OrthoFinder results, while sufficient to measure global trends such as average gene family size across all species, appeared to be insufficient to explain differences at the level of individual ROS-related gene families.

Reviewer 2 Report

Manuscript “Comparative genomics of extremophile Eukaryotes reveal plant-specific ROS network genes” by Lyall et al., presents an attempt to compare the presence and abundance of ROS-related genes across a broad array of species with a special emphasis on those which are tolerant to extreme conditions. As a result the authors were able to identify the core set of ROS scavengers and point out the differences between the kingdoms. Further, authors found that APXs, MDHARs and TRs are plant specific. Finally, with the exception of few species, no correlation between the expansion of ROS-related genes and tolerance to extreme conditions has been identified. The paper is interesting, however, from my point of view, it lacks the aspect of novelty. I also miss the interpretation of the data generated by the authors – at present, the manuscript reads like a list of observations and I do not see the major conclusions of this study. If I were the authors I would attempt to extract a few original key findings of this work and rewrite the paper to focus on discussing their functional significance in  way that would be accessible to the broad audience. Finally, the paper could benefit from an extensive fact-checking. Therefore, at present I am not able to recommend this manuscript for publication.

Major remarks:

  1. In my opinion the title does not reflect the contents of the paper. The authors promise the identification of plant-specific ROS-related genes e.g. APXs and MDHARs but is this finding really novel? According to me it is not, I see earlier works stating the same e.g. https://www.ncbi.nlm.nih.gov/pmc/articles/PMC6191929/ and papers cited therein. The presence of two classes of thioredoxin reductases is also not a new discovery. Why are these findings hinted at in the title and further emphasized in the abstract of this paper? Please consider focusing on the novel findings stemming from your work.
  2. Consider discussing the biological relevance of novel findings presented in this paper.

Minor remarks:

  1. Abstract and the rest of the paper: do not capitalize the names of groups of enzymes. This nomenclature is correct for e.g. Arabidopsis proteins, and only if there is just one protein that is being referred to.
  2. Line 36: differing => varying
  3. Line 42: molecule => molecules
  4. Line 45-48: Ascorbate and glutathione are indeed the major cellular redox buffers, and react directly with some forms of ROS e.g. superoxide. However, their direct ROS scavenging capacity towards e.g. H2O2 is negligible when compared to the reactions catalyzed by dedicated enzymes e.g. ascorbate peroxidases (plants) and glutathione peroxidases (animals) that use AsA and GSH as electron donors. If AsA was indeed a good general ROS scavenger then mutants of e.g. apx would not have any phenotypes, which is not true. Furthermore, at physiological pH, the -SH group of GSH (pKa 9.0) is protonated and not so reactive. Please revise this text to reflect the major role of GSH and ASC as enzyme cofactors.
  5. Line 54-61: This text needs revision. Clearly state which enzymes are ROS scavengers (and what is their cofactor) and which regenerate GSH and ASC. One should also remember that in plants the reduced ferredoxin can directly regenerate ascorbate. Peroxiredoxins are reduced mainly by thioredoxin/gluaterdoxin system. Further, at least in Arabidopsis, the GPX proteins should be referred to as GPX-like proteins see https://onlinelibrary.wiley.com/doi/full/10.1111/pce.12919 for reasoning.
  6. Table 1: Peroxiredoxin catalytic cycle involves thiol/disulfide exchange see e.g. https://www.ncbi.nlm.nih.gov/pmc/articles/PMC5806080/ I would not group them with peroxidases
  7. Table 1: In my version of the manuscript, majority of the references from Table 1 are not in the reference list, therefore at present I am not able to assess the quality of this section of the paper.
  8. Line 254: I think authors mean peroxisomes, not mitochondria
  9. Figure 3, 4 and 6 are too small and impossible to interpret, they should probably be arranged vertically.
  10. Line 336: homeiostasis -> homeostasis, please correct typos throughout the manuscript.

Author Response

Major remarks:

In my opinion the title does not reflect the contents of the paper. The authors promise the identification of plant-specific ROS-related genes e.g. APXs and MDHARs but is this finding really novel? According to me it is not, I see earlier works stating the same e.g. https://www.ncbi.nlm.nih.gov/pmc/articles/PMC6191929/ and papers cited therein. The presence of two classes of thioredoxin reductases is also not a new discovery. Why are these findings hinted at in the title and further emphasized in the abstract of this paper? Please consider focusing on the novel findings stemming from your work.

The focus of the paper (notably the Title, Abstract and Discussion) has been redirected towards the novel aspects of our results, particularly as they relate to extremotolerance in general.

Consider discussing the biological relevance of novel findings presented in this paper.

In accordance with the point above, the Discussion is expanded and refocused around our novel results in the context of extremotolerance.

Minor remarks:

1. Line 45-48: Ascorbate and glutathione are indeed the major cellular redox buffers, and react directly with some forms of ROS e.g. superoxide. However, their direct ROS scavenging capacity towards e.g. H2O2 is negligible when compared to the reactions catalyzed by dedicated enzymes e.g. ascorbate peroxidases (plants) and glutathione peroxidases (animals) that use AsA and GSH as electron donors. If AsA was indeed a good general ROS scavenger then mutants of e.g. apx would not have any phenotypes, which is not true. Furthermore, at physiological pH, the -SH group of GSH (pKa 9.0) is protonated and not so reactive. Please revise this text to reflect the major role of GSH and ASC as enzyme cofactors.

This addition has been made, as well as an additional figure (Figure 1 in the new draft) which highlights the GSH and GSH-AsA cycles in more detail.

2. Line 54-61: This text needs revision. Clearly state which enzymes are ROS scavengers (and what is their cofactor) and which regenerate GSH and ASC. One should also remember that in plants the reduced ferredoxin can directly regenerate ascorbate. Peroxiredoxins are reduced mainly by thioredoxin/gluaterdoxin system. Further, at least in Arabidopsis, the GPX proteins should be referred to as GPX-like proteins see https://onlinelibrary.wiley.com/doi/full/10.1111/pce.12919 for reasoning.

This reasoning is valid. However, rather than rename all such gene families for each species, we have included the following note in the Results to highlight this issue:

“For the sake of this study, putative homologs are identified using the common names for the gene family (e.g. NOX, GPX). However, as these genes are identified by homology alone, there is no guarantee that they are functional or that the method of action is the same as those typically found in model organisms.”

3. Table 1: Peroxiredoxin catalytic cycle involves thiol/disulfide exchange see e.g. https://www.ncbi.nlm.nih.gov/pmc/articles/PMC5806080/ I would not group them with peroxidases

The summary for peroxidases in Table 1 has been modified to highlight the fact that the peroxidase enzyme family can catalyse reactions involving H2O2 and other radical peroxides and can use multiple cofactors, including heme and cysteine/selenocysteine. Peroxiredoxins are thiol peroxidase enzymes, as stated in the abstract of the linked article.

4. Table 1: In my version of the manuscript, majority of the references from Table 1 are not in the reference list, therefore at present I am not able to assess the quality of this section of the paper.

This was an oversight – the citations were manually added and so did not appear in the reference manager bibliography. They are now cited in the table directly.

5. Abstract and the rest of the paper: do not capitalize the names of groups of enzymes. This nomenclature is correct for e.g. Arabidopsis proteins, and only if there is just one protein that is being referred to.

6. Line 36: differing => varying

7. Line 42: molecule => molecules

8. Line 254: I think authors mean peroxisomes, not mitochondria

9. Figure 3, 4 and 6 are too small and impossible to interpret, they should probably be arranged vertically.

10. Line 336: homeiostasis -> homeostasis, please correct typos throughout the manuscript.

All of the above minor corrections (points 5-10) have been made.

Round 2

Reviewer 2 Report

The introduced changes significantly improved the quality of this manuscript, however, this work can benefit from additional corrections:

Line 22: I suggest to remove “thioredoxin reductases” or to specify the isoforms/molecular weight

Line 59: “scavenge ROS” -> “scavenge specific forms of ROS”, this is not a general function

Figure 1: Additionally, in plants, chloroplast thioredoxins are reduced by ferredoxin-dependent thioredoxin reductase (FTR)

Line 142: specify what the abbreviation “GLR” means

Correct the remaining typos throughout the manuscript e.g. line 477 “TRRs” and line 449 “high-MR TRs”

Line 712-715: Please move this text to the first paragraph of the result section. Additionally, it should further be noted that the mere absence of an orthologue does not mean that the reaction in question is absent. It simply might be that other non-homologous enzymes are responsible for catalyzing it, case in point being the absence of DHAR in humans –there are other enzymes capable of reducing DHA see e.g. https://www.sciencedirect.com/science/article/pii/S0022283612003439

Author Response

Line 22: I suggest to remove “thioredoxin reductases” or to specify the isoforms/molecular weight

The example of “thioredoxin reductases” has been removed from this sentence.

Line 59: “scavenge ROS” -> “scavenge specific forms of ROS”, this is not a general function

Corrected.

Figure 1: Additionally, in plants, chloroplast thioredoxins are reduced by ferredoxin-dependent thioredoxin reductase (FTR)

This has been clarified in the figure legend:

“Thioredoxin is regenerated by the related TRX-specific reductase (TR). In plant chloroplasts, TRX is also reduced by a ferredoxin-dependent TR (FTR).

Line 142: specify what the abbreviation “GLR” means

This was a misspelling and has been corrected.

Correct the remaining typos throughout the manuscript e.g. line 477 “TRRs” and line 449 “high-MR TRs”

The given types have been corrected, as well as a handful of others.

Line 712-715: Please move this text to the first paragraph of the result section. Additionally, it should further be noted that the mere absence of an orthologue does not mean that the reaction in question is absent. It simply might be that other non-homologous enzymes are responsible for catalyzing it, case in point being the absence of DHAR in humans –there are other enzymes capable of reducing DHA see e.g. https://www.sciencedirect.com/science/article/pii/S0022283612003439

The text has been moved to the first paragraph of the Results, and has been expanded as below:

“For the sake of this study, putative homologs are identified using the common names for the gene family (e.g. NOX, GPX). However, as these genes are identified by homology alone, there is no guarantee that they are functional or that the method of action is the same as those typically found in model organisms. In addition, the absence of an orthologue related to a chemical pathway in an organism does not necessarily imply that that organism is unable to catalyze that reaction - it could instead be achieved through species- or lineage-specific homologues or redundant mechanisms.”